# AbsInt-AI: Agentic Heap Abstractions for Abstract Interpretation

## Abstract

Static program analysis is a foundational technique in software engineering for reasoning about program behavior. Traditional static analysis algorithms model programs as logical systems with well-defined semantics, but rely on uniform, hard-coded heap abstractions. This limits their precision and flexibility, especially in dynamic languages like JavaScript, where heap structures are heterogeneous and difficult to analyze statically. In this work, we introduce ABSINT-AI, a language-model-guided static analysis framework that augments abstract interpretation with adaptive, per-object heap abstractions for Javascript. This enables the analysis to leverage high-level cues, such as naming conventions and access patterns, without requiring brittle, hand-engineered heuristics. Importantly, the LM agent operates within a bounded interface and never directly manipulates program state, preserving the soundness guarantees of abstract interpretation. To evaluate our approach, we focus on a soundness-critical task: determining whether object property accesses may result in undefined or null dereferences. This task directly models a common requirement in compiler optimizations, where proving that an access is safe enables the removal of dynamic checks or simplifies code motion. On this task, ABSINT-AI reduces false positives by up to 34% compared to traditional static analyses with fixed heap abstractions, while preserving formal guarantees. Our ablations show that the LM's ability to interact agentically with the analysis environment is crucial, outperforming non-agentic LM predictions by 25%.

## 1 Introduction

As dynamic languages like JavaScript find their way into more backend applications with strong performance requirements, there has been a growing interest in compiling them down to more optimal forms (ang; Serrano, 2022; Chandra et al., 2016). An important obstacle for these approaches is the difficulty of performing sound static program analysis on these languages due to their dynamic behavior and extensive use of complex heap allocated data (Feldthaus et al., 2013; Antal et al., 2023; Sridharan et al., 2012). This is a problem because sound analysis is an essential element of compiler optimization (Hind, 2001; Schneck, 1973). Soundness ensures that the analysis captures all possible runtime behaviors of the program; without it, compilers cannot guarantee the safety of specific transformations.

A key challenge in sound and scalable static analysis for JavaScript is reasoning about the heap. JavaScript's dynamic object model allows programs to construct and mutate objects with unpredictable shapes, runtime-dependent fields, and implicit behavior tied to values stored within fields. Consider a typical loop that allocates multiple heterogeneous objects: some are short-lived wrappers, others are stable configuration records, and others may exhibit role-dependent behaviors encoded in field values. Traditional static analyses typically rely on uniform abstraction strategies, and often result in excessive over-approximation and imprecision. Constructing precise yet scalable heap abstractions is a major challenge for JavaScript due to its lack of static types and its permissive object model, and it remains a major bottleneck for static analysis frameworks.

In this paper, we introduce ABSINT-AI, an agentic framework that assists static analysis by performing heap abstractions. Our approach preserves the strong guarantees provided by traditional static analysis techniques while addressing some of their major limitations. Static analysis techniques analyze programs by treating them as sets of logical statements with well-defined semantics (Cousot & Cousot, 1977). This type of analysis can provide guarantees of soundness, but these methods leave

out a lot of information, such as variable names, comments, general programming design patterns, and background knowledge. LMs on the other hand, are able to take advantage of this information very well, but lack the robustness of traditional static analysis. For example, changing variable names has been shown to have a drastic impact on model performance (Zeng et al., 2022; Srikant et al., 2021). ABSINT-AI combines the best of both worlds by using LMs to provide background information to a static analyzer without losing soundness guarantees.

The key design choice in ABSINT-AI is that it preserves the formal soundness guarantee of symbolic program analysis by constraining the LM to only choose from a pre-determined set of *sound abstraction strategies* and decide *where* to apply abstractions. As a result, ABSINT-AI bounds the (inevitable) LM errors to only increased false positives (due to the aggressive abstraction decision) or slow down the convergence of the analysis (reduce to the precise but expensive analysis) without compromising the soundness.

Specifically, ABSINT-AI consists of a custom static analysis pipeline that invokes an agentic LM framework at key decision points - most notably before fixpoint computations in unbounded loops, where the choice of abstraction heavily influences convergence and precision. At each such point, the agent inspects the current analysis state, including the heap, code, and abstraction history. Based on this inspection, it selects appropriate abstraction strategies for each allocation site, such as merging objects using recency-abstraction, field sets, or value similarity. If the available information is insufficient to make a confident decision, the agent can request additional targeted analysis by executing the loop body for more iterations to refine its understanding. This interactive, goal-directed behavior enables adaptive, context-sensitive abstraction decisions and also allows the abstractions themselves to reflect higher-level semantic concepts. For example, if objects contain a `role` field, the agent can select a value-sensitive abstraction that merges all "teachers" into one object and all "students" into another, allowing domain-specific concepts to guide the abstractions themselves.

We evaluate our approach on the downstream task of detecting accesses to non-existent object fields, a common source of runtime errors in JavaScript. We compare our system against WALA (Santos & Dolby, 2022) and TAJS (Jensen et al., 2009), two state-of-the-art static analysis frameworks that are representative of conventional heap abstraction strategies. Our evaluation of real-world JavaScript programs shows that ABSINT-AI achieves up to a 34% reduction in false positives while maintaining soundness. Our ablations show that this improvement stems not just from more expressive abstractions, but from the agent's ability to interact with the analysis and adapt its choices to the program context. When run with fixed symbolic abstractions or using the LM in a single-shot, non-interactive mode, the false positive rate increases by 88% and 25%, respectively. These results highlight the benefit of adaptive, semantically informed heap abstractions in improving the practical effectiveness of sound JavaScript analysis.

## 2 MOTIVATING EXAMPLE

Static analyses rely on heap abstractions (summaries of sets of objects), to reason about dynamic, heap-manipulating programs. The precision of these abstractions has a huge impact: too coarse and the analysis produces spurious warnings; too fine and it may never converge.

Modern JavaScript programs often construct diverse heap objects with different structural patterns and semantic roles, even within the same control-flow context. A one-size-fits-all heap abstraction applied uniformly across the entire program can lead to loss of precision or unnecessary state explosion. Consider the example in Figure 1, where each iteration of processElements allocates two distinct objects: a short-lived wrapper (`box`), and a structured configuration object (`config`). Each of these demands a different abstraction strategy. For instance, box can be aggressively summarized without affecting ~~soundness~~precision, while config exhibits a fixed field structure where only a single field, `valid`, must remain precise for correct downstream control flow. While it is theoretically possible to hand-engineer heuristics that assign abstraction strategies based on object structure or access patterns, doing so at scale quickly becomes brittle, complex, and difficult to maintain. To the best of our knowledge, existing analyses do not adapt their heap abstractions per object, due to the complexity and brittleness of manually encoding such decisions.

However, many real-world objects contain semantic hints in field names or surrounding code that indicate how they should be abstracted. For example, the field `valid` suggests that the `config` object

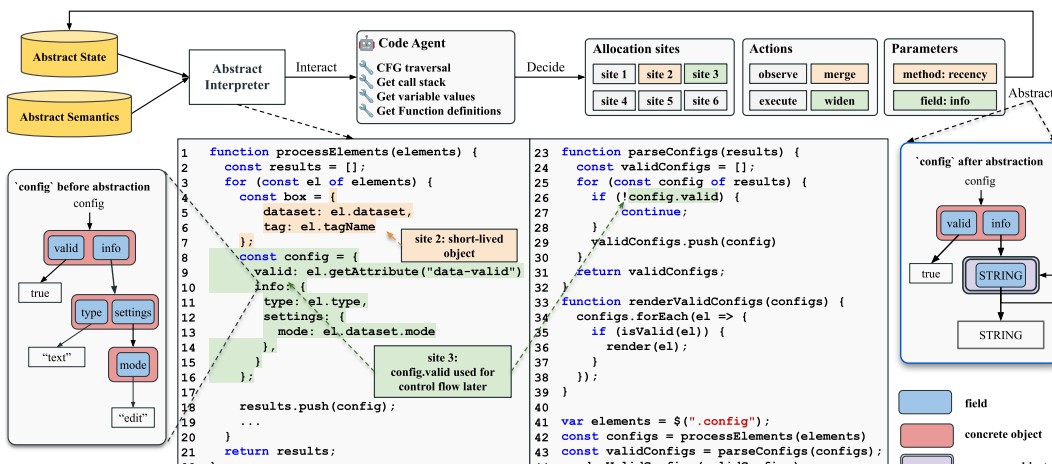

Figure 1: When ABSINT-AI encounters an unbounded loop, it suspends analysis and interacts with the language model agent for abstraction decisions. The agent selects a recency abstraction for `box` and a field-based widening for the `info` field of `config`, preserving relevant structure while ensuring convergence. A concrete instance of `config` is shown on the left, with its abstracted form on the right. These per-allocation-site abstraction decisions guide the analysis to a sound fixpoint.

encodes access control logic, which is later reflected in a guard on `config.valid`. These high-level concepts such as "valid" configurations are difficult to capture using purely syntactic heuristics or static types, but are easily interpretable by language models. An agentic abstraction strategy can leverage such semantic cues to select more appropriate abstractions: preserving distinctions between roles, ~~merging only safe-to-abstract fields, or even proposing domain-informed widenings~~or widening only fields relevant to the analysis domain. This enables adaptive precision where it matters, and aggressive summarization where it doesn't—leading to more efficient and accurate analyses.

In ABSINT-AI, a language model acts as an agent that guides heap abstraction dynamically over the course of the analysis. Returning to the example in Figure 1, the agent might decide to apply recency abstraction to the short-lived `box` object and a field-set abstraction to the structured `config` object (preserving only `config.valid`). These decisions are not hardcoded: the agent queries the analysis environment for relevant context (such as variable values and function definitions), and may request additional loop iterations to test its abstraction choices. Crucially, all semantics and state transitions are handled by a traditional abstract interpreter, ensuring that soundness is preserved. The agent's role is purely to steer how the heap is abstracted, enabling more precise and efficient analysis by tailoring abstraction to the semantics of the program.

## 3 METHODOLOGY

ABSINT-AI is based on traditional abstract interpretation, but queries an LM to decide how to merge summary nodes at key points in the analysis. The workflow of ABSINT-AI can be found in Figure 1.

### 3.1 BACKGROUND

**Static program analysis**. Static program analysis aims to reason about all possible executions of a program. A key property is *soundness*, meaning the analysis never misses a real bug (no false negatives). The tradeoff is *precision*: overly coarse reasoning introduces spurious warnings (false positives).

To ensure scalability, analyses use abstractions that merge unbounded program behaviors (e.g., integers, heap objects) into finite summaries. For heap-manipulating languages like JavaScript, this typically means summarizing many concrete objects into a smaller set of abstract objects. The challenge is choosing what to merge: aggressive abstraction hurts precision, while conservative abstraction may prevent convergence. Prior work (Kanvar & Khedker, 2016) has developed many hand-written heuristics for heap abstractions. Our approach replaces such heuristics with LM-guided, ~~context-sensitive~~ adaptive abstractions.

**Abstract interpretation**. Abstract interpretation (Cousot & Cousot, 1977) soundly approximates program behavior by tracking an abstract state that summarizes all possible concrete states. Each program operation updates the abstract state according to sound rules; for loops, iterative application yields a fixpoint that safely over-approximates all executions. For heap-manipulating programs, this requires a heap abstraction that merges potentially unbounded sets of objects into finite summary objects (Sagiv et al., 1998; Kanvar & Khedker, 2016). Traditional analyses rely on hand-crafted heuristics for when and how to introduce summaries. Our work instead uses a language model to guide these choices adaptively. (We provide a more detailed overview of abstract interpretation and heap abstractions in Appendix A.)

## 3.2 ABSTRACT INTERPRETATION

Abstract interpretation requires an abstract domain as well as modeling of the heap. In this section, we briefly describe our abstract domain, our two-level representation of the heap, and when we invoke the LM for summarization. The full analysis supports prototypal inheritance, recursion, loops, and closures. Additional details can be found in the appendix.

**Abstract Domain.** Our abstract domain keeps track of heap objects using concrete nodes and summary nodes. Summary nodes represent a set of possible concrete nodes.

Each node is a dictionary from primitive or abstract values to other values. Our domain of primitive values is based off of TAJS (Jensen et al., 2009), one of the first abstract interpretation based analyses for Javascript. The abstract domain and transfer functions are fixed; the LM agent does not alter the semantics of the analysis. Its role is limited to guiding when and where widening and merging operators are applied. Additional details on our abstract domain can be found in the appendix. The most important runtime decision of ABSINT-AI is deciding when summarize heap nodes. We keep two separate heap structures, referred to as the local heap and global heap.

```
1   var global = 0;
2   var global_obj = {};
3   function inc_global() {
4       let obj = {f: 1};
5       obj.f += 1;
6       global = global + obj.f;
7   }
8   function access_obj() {
9       if (global > 10) {
10          var f = global_obj.foo.bar; // bug
11      }
12  }
13  var btn1 = document.createElement("button");
14  var btn2 = document.createElement("button");
15  btn1.addEventListener("click", inc_global);
16  btn2.addEventListener("click", access_obj);
```

Figure 2: `inc_global` needs to be run at least 10 times before the bug on line 11 is triggered.

**Local heap.** The local heap is used for precise representation for objects within local procedures, such as a local object allocation in a function call. It is flow-sensitive (Kildall, 1973), taking into account the order of statements. For example, in Figure 2, `obj` on line 4 is tracked in the local heap.

**Global heap.** The global heap is a much less precise representation for objects that are accessed and manipulated by multiple functions. The global heap captures all possible relationships between globally visible objects at any point in the execution. The global heap is motivated by flow-insensitive analysis (Weihl, 1980; Cousot & Cousot, 1977). This has two benefits: (1) It is much cheaper, as we don't have to keep track of a separate heap for each program location, and (2) it allows different functions to be analyzed independently; the global heap considers all the possible heap states at the point when the function is invoked, and the analysis of the function can reveal if any additional relationships need to be added to the global heap. Summarization only happens in the global heap.

We draw a distinction between the local and global heap because JavaScript programs tend to be reactive, with execution driven largely by external events. This has important implications for analysis, as the analysis can't assume the program will simply execute starting at the beginning from a well defined initial state. Take the example in Figure 2, where `inc_global` is invoked by an event handler and must be executed at least 10 times in order to trigger the bug on line 11. Keeping two separate heaps allows us to to track global dependencies while not losing precision for local procedures.

**Agent Invocation.** A key challenge in abstract interpretation is to reach a fixpoint without losing too much precision when analyzing potentially unbounded loops. Because fixpoint computation requires merging abstract states across iterations, the choice of how to abstract heap objects allocated within the loop has a direct impact on both the precision and termination of the analysis.

Take the example in Figure 1. There are two objects, `box` and `config`. Each loop iteration allocates two objects: `box`, which is short-lived and well-suited to recency abstraction, and `config`, which contains a critical field (`valid`) that must remain precise. A uniform abstraction by allocation site would collapse these distinctions, introducing spurious behaviors. ABSINT-AI addresses this by invoking the LM agent at unbounded loops to choose abstraction strategies per object, balancing semantic precision with soundness and convergence. The agent is only invoked at unbounded loop joins, not at if–then–else merge points. Conditional branches use standard abstract joins and do not require agent intervention.

### 3.3 Agentic Heap Abstractions

The agent in our framework serves as an interactive component embedded within the analysis loop. Its role is to select heap abstraction strategies, but unlike a static classifier, it behaves as an agent that operates under partial information and interacts with its environment to gather context before acting.

The agent is not invoked as a one-shot oracle. Instead, it operates as a environment-interacting agent that gathers information over time. To make informed abstraction decisions, the agent interacts with the abstract interpreter and the abstract state to selectively gather semantic information from the program. Rather than exposing the entire program or heap state, which would overwhelm the agent and obscure the relevant context, we treat the interpreter as a queryable environment. This avoids a common challenge in machine learning for code: programs often contain far more information than an LLM can meaningfully process, especially in settings with deep heap structure.

The agent's outputs are limited to a predefined set of sound abstraction strategies, and it never directly manipulates program state or executes code. The underlying abstract interpreter remains responsible for all semantic computation and fixpoint reasoning. This architectural separation allows us to embed an adaptive, learning-driven agent within a sound static analysis framework—enabling high-level decision-making informed by context and semantics, while preserving formal correctness guarantees.

**Agent Interaction**. The agent is initialized with the current abstract state, including visible variables, relevant allocation site data, and any previously encountered heap shapes. It then enters an interactive decision-making loop. During this loop, the agent can issue queries to the abstract state for more information, such as requesting variable values, inspecting function definitions, or examining the heap shape. If the available information is insufficient, the agent may also postpone its decision making by requesting additional abstract loop iterations, allowing it to observe how the heap evolves over time. This enables the agent to defer commitment while gathering contextual evidence. We experimented with providing the full program and abstract state directly in the

---

**Algorithm 1** Agentic Heap Abstraction Algorithm

**Require:** Loop $\mathcal{L}$, Analysis state $\mathcal{S}$, Allocation Sites $\mathcal{A}$
1: $b \leftarrow 0$ {Interaction counter (queries + executions)}
2: $\mathcal{A}' = \text{NONE}$
3: **while** $b < $ budget **do**
4:     Agent selects action $a \in \{\text{INFO}, \text{EXEC}, \text{SELECT}\}$
5:     **if** $a = \text{INFO}$ **then**
6:         Agent queries $\mathcal{S}$ for program information
7:         $b \leftarrow b + 1$
8:     **else if** $a = \text{EXEC}$ **then**
9:         Abstract Interpreter executes one iteration of the loop
10:         $b \leftarrow b + 1$
11:         continue
12:     **else if** $a = \text{SELECT}$ **then**
13:         Agent selects sites $\mathcal{A}' \subseteq \mathcal{A}$ to abstract
14:         break
15:     **end if**
16: **end while**
17: **if** $\mathcal{A}' = \text{NONE}$ **then**
18:     Agent selects $\mathcal{A}' \subseteq \mathcal{A}$ to abstract
19: **end if**
20: **for** $a_i \in \mathcal{A}'$ **do**
21:     Agent selects (Strategy,Parameters)
22:     Updated mapping in $\mathcal{S}$ from $a_i$ to strategy for $\mathcal{L}$
23: **end for**

prompt, but the abstract heap often exceeded the model's context window for larger or deeply nested programs. To ensure stable, reproducible behavior, the agent instead accesses information incrementally through `INFO` queries, retrieving only the specific variable or function summaries needed for each decision.

The interaction is bounded: the agent operates under a fixed query and iteration budget to ensure termination. Once satisfied, the agent returns a set of abstraction directives, specifying how the interpreter should merge and widen objects associated with each allocation site. The interpreter then executes the loop abstracting the heap as directed by the agent. If the abstract state does not reach a fixpoint within five iterations, it re-queries the agent for new abstraction strategies. Algorithm 1 contains a detailed description of our procedure.

The agent performs two decision stages:

**1. Selecting which allocation sites to summarize.**

- At each loop iteration, the interpreter identifies allocation sites whose abstract states changed.
- The agent receives a prompt containing:
  - The loop body and relevant code snippet.
  - A summary of changed allocation sites (object structures, points-to sets).
- The agent can issue a number of **information gathering** requests (for example, querying the current abstract value of a variable or requesting a summary of a function's behavior) or a simulated loop execution with the abstract interpreter.
- Finally, the agent selects a set of allocation sites to summarize or merge, ensuring convergence before the next iteration.

**2. Choosing a merging strategy and widening strategy for each selected site.**

- For every selected allocation site, the interpreter asks the agent to choose one of the predefined parameterizable **merging strategies** for that site.
- The agent picks among them using natural-language cues from code and variable names.
- After picking a merging strategy, the interpreter asks the agent to choose one of the predefined parameterizable **widening strategies**.

Every action the agent can take is predefined, finite, and sound—it cannot invent new abstractions, only select among existing ones—and all interactions are deterministic within the interpreter.

**Information Gathering**. The agent gathers information through a small set of read-only queries to ABSINT-AI:

- **Variable inspection**: Requesting abstract values of in-scope variables.
- **Function introspection**: Retrieving the definition of local functions in scope.
- **Loop execution**: Requesting additional iterations with the abstract interpreter to observe how heap structures evolve.

These interactions allow the agent to incrementally reduce uncertainty and focus attention on semantically meaningful heap behaviors without drastically increasing the input size. In particular, loop execution supports deliberate abstraction delay, giving the agent a richer view of program dynamics before committing to a strategy.

~~Abstraction decisions.~~ **Merging Strategies.** Once the agent has identified which allocation sites require abstraction, it selects a merging strategy for each. This determines how objects allocated at that site are grouped during join operations. The agent chooses from the following predefined strategies:

- **Allocation-site merge**: Collapses all objects created at the same program location into a single abstract object.
- **Recency merge**: Preserves the most recently allocated object at that site; merges older instances.
- **Field-sensitive merge**: Groups objects with the same fields.
- **Role-based merge**: Partitions objects based on semantically meaningful field values (e.g., role), allowing distinctions like "student" vs. "teacher" to be preserved.

In particular, role-based merging requires semantic understanding of field names and value meanings; it is very difficult to implement role-based merging using purely symbolic techniques. Identifying that a specific field should guide abstraction boundaries is often a decision that depends on natural language cues and program intent.

After selecting a merging strategy for an allocation site, the agent also specifies a widening strategy. Widening determines how abstract heap objects are generalized over time as they are revisited across loop iterations. The agent chooses from the following strategies:

- **Field-set widening**: widen a selected subset of fields, leave the others concrete.
- **Field merging**: Merge the fields together, and select another widening strategy for the values. This is for handling infinitely growing objects.
- **Full widening**: recursively widen the entire object into a single shape.
- **Depth-based widening**: Collapse structures beyond a fixed depth threshold

These strategies allow the agent to control the granularity of abstraction per object: preserving precise structure where it matters while widening aggressively in parts of the heap that are less semantically relevant. As with merging, widening strategies are selected per allocation site and parameterized to balance precision with scalability.

## 3.4 DOWNSTREAM TASK

As a downstream task to test the precision of ABSINT-AI, we detect the following situations (1) accessing a property of `null` or `undefined` and (2) reading an absent property of an object.

Abstracting unnecessarily can lead to false positives. Take the example in Figure 3. If `userId` on line 1 gets abstracted to the abstract NUMBER type, then the object access on line 3 is reported as a possible read of an absent property. `userId` could take the value of all possible numbers, but `names` only has the the property `100`.

```
1  let userId = 100; // abstracted to NUMBER.
2  let names = {100: "Jane"};
3  names[userId]; // False positive
```

Figure 3: False positive due to `userId` getting abstracted to the abstract NUMBER type.

**Intersection of multiple runs.** Different abstraction choices in a program can lead to different sets of reported bugs. For example, when analyzing the program in Figure 3, ABSINT-AI may choose to abstract the `userId` field in some runs but leave it concrete in others. This variation can affect which false positives are reported. However, because each run is individually sound, any bug that does not appear in *any* run is guaranteed not to be real. This allows us to improve precision by taking the intersection of reported bugs across multiple runs (similar in spirit to self-consistency approaches (Wang et al., 2022b)) while preserving full soundness.

## 4 EVALUATION

Our evaluation focuses on two key questions: (1) How does our system perform compared to existing static analysis tools? (2) How important is agentic decision-making relative to fixed symbolic strategies or direct LLM prediction? To answer these, we compare against two established baselines (TAJS and WALA), conduct targeted ablations isolating the role of the agent, and present a case study demonstrating the system's ability to preserve meaningful heap structure.

## 4.1 BASELINES

**TAJS**. TAJS (Type Analysis for JavaScript) is a performs flow-sensitive, context-sensitive, and partially path-sensitive static analyzer designed for sound and scalable analysis of JavaScript programs Jensen et al. (2009). TAJS is based on abstract interpretation, including specialized heap abstractions such as allocation-site abstraction and recency abstraction, to model JavaScript's dynamic object behavior.

**WALA**. WALA (T. J. Watson Libraries for Analysis) is a general-purpose static analysis framework that supports multiple languages, including JavaScript Santos & Dolby (2022). Unlike TAJS, WALA is

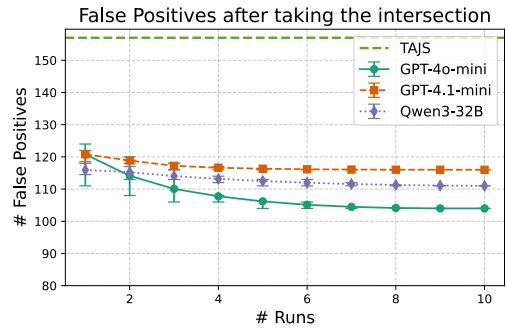

Figure 4: Running multiple times and taking the intersection of the reported bugs allows us to improve precision while maintaining soundness.

Table 1: Overall mean performance across the Dataset. #FP stands for False Positives. Fewer is better.

| | Model | # FP↓ | % Improve |
|---|---|---|---|
| Baselines | TAJS | 157 | 0% |
| | WALA | 312 | -98.7% |
| | Symbolic ABSINT-AI | 220 | -28.6% |
| Mean | GPT-4o-mini | 125 | 20.4% |
| | GPT-4.1-mini | 127 | 19.1% |
| | Qwen3-32B | 117 | 25.5% |
| Intersection | GPT-4o-mini | 104 | 33.7% |
| | GPT-4.1-mini | 116 | 26.1% |
| | Qwen3-32B | 111 | 29.0% |
| | Full Intersection | 97 | 38.2% |

not based on abstract interpretation and performs flow-insensitive heap analysis, using a combination of allocation-site abstraction and context-sensitive pointer analysis.

**Symbolic ABSINT-AI**. We also include a baseline that runs ABSINT-AI using a fixed abstraction configuration without LM guidance. This baseline selects a conservative widening strategy across all allocation sites, simulating how our analysis would perform without agentic control. It serves to isolate the contribution of the LM-driven adaptivity from the underlying analysis framework. Symbolic ABSINT-AI begins with recency-based merging and a depth-1 field-sensitive abstraction. If the loop fails to converge within 50 iterations, it switches to widening the entire object while maintaining recency-based merging. If convergence still fails after another 50 iterations, it falls back to a fully allocation-site-based abstraction.

**Dataset.** To evaluate our approach, we curated a benchmark of 17 self-contained JavaScript programs from the Big Code dataset Raychev et al. (2016), the V8 benchmark suite, and Github. We filtered for programs that were self-contained and did not use builtins excessively, as this greatly increases the imprecision of the analysis (`Math.floor`, for example, requires modeling the `Math` library to analyze precisely). These require substantial modeling effort and introduce orthogonal complexity. We also excluded object-oriented programs that rely too heavily on classes and `let` statements, since TAJS and WALA do not support Javascript features after ES2015. For context, prior work such as TAJS evaluated on 8 programs (Jensen et al., 2009), underscoring the difficulty of assembling larger benchmarks for sound JavaScript analysis. All 17 benchmarks were manually inspected to confirm that the property of interest (absence of unsafe property accesses) holds. A detailed description of the dataset can be found in the Appendix.

## 4.2 PERFORMANCE

We evaluate ABSINT-AI using three different language models: GPT-4o-mini, GPT-4.1-mini, and Qwen3-32B. To compare against TAJS and WALA, we measure the number of (1) possible accesses to a property of `null` or `undefined` or (2) possible reads of an absent property of an object. In this setting, lower values indicate greater precision, reflecting fewer spurious results caused by imprecise heap abstraction. We run ABSINT-AI 10 times across our benchmark per model across our 17-program benchmark and report the mean results in Table 1.

Our agent-guided approach reports significantly fewer false positives than either baseline, achieving an average reduction of approximately 20%. This improvement stems from the agent's ability to select tailored abstraction strategies that avoid over-merging or premature widening, which often cause TAJS and WALA to lose key field or value distinctions.

**Intersection.** As described in Section 3.4, one benefit of maintaining soundness is that we can safely take the intersection of reported errors across multiple runs, improving precision without risking missed bugs. Figure 4 shows the effect of taking intersections across multiple runs. As expected, the language model often makes different abstraction decisions, leading to partially overlapping sets of reported warnings. By intersecting the results across multiple runs, either for a single model or across

all three, we can substantially reduce false positives. On average, intersecting runs from a single model improves precision by 8%; intersecting all 30 runs across all models yields a 13% reduction in false positives over any individual run. We find that intersecting the top 3–4 runs gives the steepest improvement, with diminishing returns after 6 runs.

**Run time.** We also compare the runtime performance of ABSINT-AI against TAJS and WALA. As expected, ABSINT-AI is slower, primarily due to our prototype implementation in Python, whereas both TAJS and WALA are written in Java. Much of the overhead comes from the interpreter itself, *not* from querying the agent. For example, when using GPT-4.1-mini, ABSINT-AI takes 500 seconds to run across our dataset, 189 of which is spent on agent interaction. Of the 500 seconds required to run across our dataset, 189 seconds correspond to agent interaction—the network latency and inference time of querying the agent—while the remaining time reflects the Python interpreter's overhead and a more detailed heap representation. When the agent is disabled in Symbolic ABSINT-AI, the analysis yields comparable precision to TAJS/WALA but remains slower. In contrast, TAJS and WALA complete their analysis in approximately 20 seconds.

## 4.3 ABLATIONS

**Ablation with symbolic abstractions.** To isolate the contribution of the agent itself, we conducted an ablation study comparing ABSINT-AI to a purely symbolic variant that uses the same abstraction strategies but without agentic selection. In this setup, the analysis starts with the most precise abstractions and applies a fixed conservative widening strategy if the loop fails to converge within 10 iterations. If the analysis still does not converge after 20 minutes, we terminate and collect any reported warnings up to that point.

Table 1 shows that this symbolic version performs significantly worse: despite failing to converge on five benchmarks, it still produces 28.6% more false positives than TAJS. This highlights that the benefit of ABSINT-AI does not come merely from using expressive abstractions, but from the agent's ability to adaptively choose when and how to apply them based on program context.

**Ablation with non-agent LLM.** To isolate the impact of agentic interaction, we compare our full system to a variant that uses the same language model, but in a non-agentic, single-shot setting. In this baseline, the model is prompted to select abstraction strategies directly, without the ability to query the interpreter, inspect intermediate state, or request additional loop iterations. This version performs consistently worse than our full system, show that the ability for the model to gather evidence and defer commitment is important for robust and context-sensitive decisions. As seen in Figure 5, the direct prediction consistently performs about 25% worse across our benchmarks.

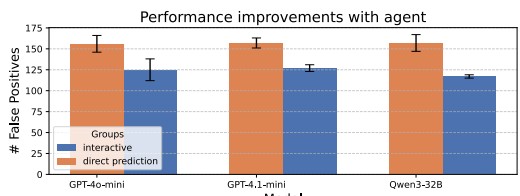

Figure 5: Performance improvements of an interactive agent vs. direct abstraction prediction.

## 4.4 CASE STUDY ON CONWAY'S GAME OF LIFE

To illustrate the benefits of agent-guided abstraction, we present a case study from our benchmark based on Conway's Game of Life in Figure 6. The `cell_state` variable represents a 3×3 grid of integers, updated over `n` iterations by the `newGeneration` function. While the contents change, the structure remains fixed across iterations; a property inherent to the game's rules. ABSINT-AI identifies that only the integer values need to be abstracted, preserving the shape of the array and producing a precise heap abstraction.

```
1  var cell_state = [
2     [0, 1, 0],
3     [0, 1, 0],
4     [0, 1, 0]
5  ]
6  var n = parseInt($("#iterations"));
7  for (var i = 0; i < n; i++) {
8     cell_state = newGeneration(cell_state);
9  }
```

Figure 6: A snippet from Conway's Game of Life.

In contrast, symbolic baselines often over-abstract the structure itself, prematurely merging array shapes and losing row-level distinctions. This highlights how the agent draws on both program syntax and semantic cues such as common data patterns to guide more precise abstraction decisions.

## 5 RELATED WORK

**LMs in program analysis.** LMs have been applied to a wide range of program analysis tasks, including type inference, fuzzing, vulnerability and resource leak detection, code summarization, and fault localisation (Peng et al., 2023; Wei et al., 2023; Wang et al., 2023b; Xia et al., 2024; Yang et al., 2023b;a; Deng et al., 2023; Mathews et al., 2024; Liu et al., 2023; Wang et al., 2023a; Mohajer et al., 2023; Cai et al., 2023; Geng et al., 2024; Ahmed et al., 2024; Wang et al., 2022a; Wu et al., 2023). However, none have been applied to static analysis while preserving soundness guarantees. More recently, several neurosymbolic approaches combine static analysis with LMs: LLift (Li et al., 2024a) filters false positives from UBITect (Zhai et al., 2020), IRIS (Li et al., 2024c) augments CodeQL (Avgustinov et al., 2016) for taint analysis, and InferROI (Wang et al., 2024) detects resource leaks in Java programs. While effective at improving precision, all of these systems sacrifice soundness once neural predictions are introduced.

**Program analysis for Javascript.** Much prior work on JavaScript analysis has focused on unsound but pragmatic tools for bug finding and security. These tools aim to detect likely vulnerabilities or errors in real-world programs, often trading soundness for scalability and precision (Li et al., 2022; Fass et al., 2019; Kang et al., 2023; Yu et al., 2023; Guo et al., 2024; Kang et al., 2025). While effective for finding particular security issues in practice, these approaches do not provide soundness guarantees. As a result, they are not suitable for many downstream tasks that depend on full program coverage, such as compiler optimizations or transformations, where missing even a single feasible behavior can invalidate correctness. Our work, by contrast, maintains the formal soundness of abstract interpretation while improving its precision via adaptive heap abstraction.

**LMs in sound reasoning.** Machine learning has been used to guide compiler optimization selection (Ansel et al., 2014; Huang et al., 2019), proof search and theorem proving (Bansal et al., 2019), as well as in program synthesis (Li et al., 2024b) and SAT/SMT solving (Ganesh et al., 2022), where learned components suggest strategies or rule orderings without affecting overall soundness. In contrast, abstract-interpretation-based program analysis forms a distinct line of work, traditionally relying solely on manually designed heuristics for abstraction and widening. To our knowledge, no prior system has incorporated large language models or other ML components into this framework while preserving soundness. Our method is the first to do so by constraining the LLM to select among a fixed, verified set of abstraction operators within a sound abstract domain.

## 6 LIMITATIONS AND CONCLUSION

**Scalability**. A limitation of ABSINT-AI is that it does not scale to large JavaScript codebases (e.g., 2,000+ lines). This is a broader issue with JavaScript static analysis: neither TAJS nor WALA converged on such programs in our experiments. The challenge stems from the dynamic and object-heavy nature of real-world JavaScript. While our agent-guided approach adds adaptivity, our prototype and reliance on whole-program analysis similarly limit scalability. Addressing this is an important direction for future work.

In this work, we propose a method to augment static analyzers with an agentic LM for heap abstractions. We present ABSINT-AI as a proof-of-concept and an evaluation showing that augmenting static analysis with LMs can have a dramatic improvement on the precision without losing soundness guarantees.

## 7 REPRODUCIBILITY STATEMENT

We have included our source code along with instructions to reproduce the experiments in the supplementary material.

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

## A BACKGROUND

**Soundness and precision.** Traditional static program analysis is often split between sound and unsound analyses. Soundness is the quality of static analyzers which guarantees that the analysis models an *over-approximation* of the target program's behavior, but may model behaviors that do not actually occur in any execution. The *precision* of the analysis is the extent to which the analysis avoids such spurious results. In short, a program analysis is *sound* if there are no false negatives. A program analysis is *precise* if there are not many false positives.

**Abstractions in static analysis.** Static analysis algorithms achieve scalability and soundness by using *abstractions* in their analysis. Programs often manipulate unbounded resources (e.g., integers, heap structures). Abstractions merge a potentially infinite set of objects into a single *summary* object to ensure convergence and for scalability. A key challenge is choosing *what* to abstract in the target program to ensure convergence while retaining as much important information as possible. There has been a rich body of literature on improving precision and scalability of heap abstractions (Kanvar & Khedker, 2016). In this work, we use an LM to decide what should be abstracted in the target program.

**Abstract Interpretation.** Abstract interpretation is a framework for analyzing programs by soundly approximating their behavior through the use of an *abstract state* that summarizes the set of possible states that a program can be in at different points in the execution (Cousot & Cousot, 1977). For simple programs manipulating scalar values, the abstract state is usually a simple mapping from variable names to abstract values representing sets of numbers. For example, an integer variable may be assigned the abstract value POSITIVE, representing all positive integers, to indicate the fact that its concrete value is guaranteed to be a positive value on any execution of the program. Abstract interpretation works by interpreting the program using rules that describe how each operation available in the language transforms the abstract state into new abstract states. For example, a rule may indicate that the addition of two POSITIVE numbers always results in a positive number. Soundness of the analysis is guaranteed by ensuring the soundness of each individual rule; for programs with loops, the analysis needs to be executed iteratively, and the theory of abstract interpretation ensures that once the abstract states converge to a fixpoint, this fixpoint will be a sound representation of the set of possible states that any execution of the program can reach.

For heap manipulating programs, the abstract state must include an abstraction of the heap which represents all the possible states of the heap a program might exhibit at a given point in time (Sagiv et al., 1998). There is an extensive literature on heap abstractions (Kanvar & Khedker, 2016), but all of them have a few elements in common. One important element is the use of *summarization* to represent multiple objects which may be living in the heap at a given point in the execution as a single *summary object*. Summarization allows the analysis to use a bounded representation for the potentially unbounded set of objects that can live on the heap on any arbitrary execution. Traditional abstract interpretation frameworks rely on complex heuristics to determine when and how to introduce summary nodes during program analysis to allow the analysis to maintain precision while quickly converging to a reasonably sized representation of the abstract heap. Our goal for this work is to replace those heuristics with an LM which can take advantage of its background knowledge of concepts used in the code as expressed through variable names, field names and comments.

# B ABSTRACT INTERPRETATION DETAILS

## B.1 ANALYSIS DETAILS

**Functions** In Javascript, functions are stored as objects on the heap. We include a `__code__` property storing the function body to be executed. At the beginning of the analysis, ABSINT-AI scans the entire program, and generates a *schema* for each function. The schema for each function contains which variables are local to the function and which variables are accessed by other functions. We refer to variables that are local as *private*, and variables that are accessed by other functions as *shared*. Each time a function is executed, an environment is initialized according to the schema for that function. When a function is defined, is initialized with a `__hf__` field set to the current heap frame. The `__hf__` field is used to model scopes and closures. When the function returns, the stack frame $\sigma$ is popped from the stack, and the stack pointer is decremented.

**Scopes and Closures** Whenever a function is called, a new stack frame $\sigma$ is pushed, along with a corresponding heap frame. The stack pointer for the current stack frame is updated to point to $\sigma$. The private variables for that function are stored in the stack frame $\sigma$, and any shared variables are stored in the heap frame. The heap frame is initialized with a parent field `__parent__` which is used to model the scope chain. The `__parent__` field points to the `__hf__` field for the function being initialized.

To lookup a variable name in the environment, ABSINT-AI first checks the current stack frame. If it finds a value for the variable, it returns the value. If it doesn't, it checks the corresponding heap frame for the stack frame, and then follows the chain of `__parent__` pointers until it finds the variable.

**Recursion** ABSINT-AI keeps track of all functions that have been called but have not finished executing yet. Whenever it encounters a recursive call, ABSINT-AI sets the return value to a recursive placeholder and stores a hash of the function that is called. When the function returns, ABSINT-AI checks the return values and any allocated heap objects for recursive placeholders for the function and fills them in with the return values.

## B.2 ENVIRONMENT

In this section we describe how ABSINT-AI represents the abstract state. We define concrete and abstract values. $H_L$ refers to the concrete heap, $H_G$ refers to the global heap, and $\sigma$ refers to the stack. $\tau$ is an abstract type, $C$ refers to constants, $obj$ and $\widetilde{obj}$ refer to concrete and abstract objects. $val$ and $\widetilde{val}$ refer to the values that a variable can take.

$$
\begin{array}{rcl}
val & ::= & a \,|\, obj \,|\, \widetilde{val} \\
\widetilde{val} & ::= & C \,|\, \widetilde{a} \,|\, \tau \,|\, \widetilde{obj} \\
\tau & ::= & Bool \,|\, Null \,|\, Num \,|\, String \\
obj & ::= & \tau \to val \,|\, C \to val \\
\widetilde{obj} & ::= & \tau \to \widetilde{val} \,|\, C \to \widetilde{val} \\
H_L & ::= & a \to val \\
H_G & ::= & \widetilde{a} \to \widetilde{val} \\
\sigma & ::= & C \to val
\end{array}
$$

## B.3 SYNTAX

$$
\begin{array}{rcl}
op & ::= & + \,|\, - \,|\, \div \,|\, \cdot \,|\, ... \\
E & ::= & id \,|\, E.field \,|\, E[E] \,|\, foo(E) \,|\, E_1[E_2](E_3,E4,...) \,|\, \text{function}(x_0,x_1,...)\{S\} \\
& & |\, \text{new } foo(E_1,E_2,...) \,|\, C \,|\, \{f:E\} \\
varDef & ::= & \text{var } id = E \,|\, \text{let } id = E \,|\, \text{const } id = E \\
Stmt & ::= & varDef \,|\, id = E \,| \\
& & E.f = E \,|\, E[E] = E \,|\, \text{def } foo(x_1,x_2,...,x_n)\{Stmt\} \,| \\
& & \text{if } (E)\{Stmt\} \text{ else } \{Stmt\} \,|\, \text{class } foo\{Stmt\} \,| \\
& & \text{return } E \,|\, \text{for } (varDef; \text{ E}; \text{ Stmt})\{Stmt\} \\
& & \text{for } (varDef \text{ in E})\{Stmt\} \,|\, \text{while } (E)\{Stmt\} \,|\, Stmt;Stmt
\end{array}
$$

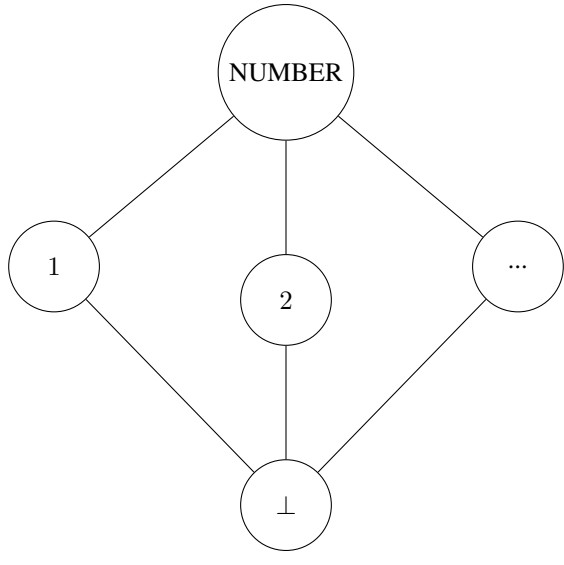

Figure 7: Number Lattice.

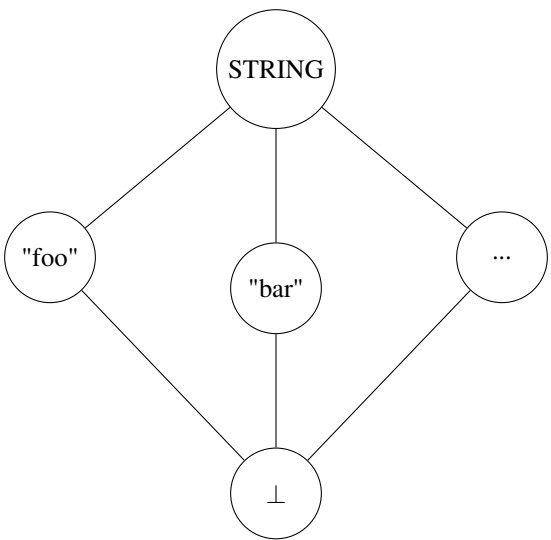

Figure 8: String Lattice.

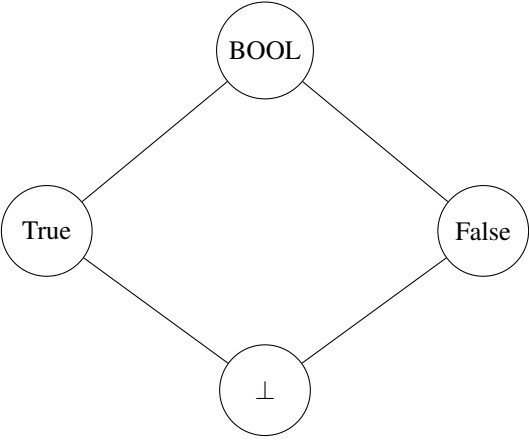

Figure 9: Boolean Lattice.

Figure 10: Null Singleton.

## B.4 SEMANTICS

### B.4.1 FUNCTIONS

This section is several functions we use, such looking up a variable name and initializing a new schema for a function.

lookup(id)
$$\frac{s \equiv \emptyset \quad \theta = \emptyset}{\langle lookup(H_L, H_G, s, id) \rightarrow \theta \rangle}$$

$$\frac{s \in H_L \quad id \in H_L(s) \quad \theta = s}{\langle lookup(H_L, H_G, s, id) \rightarrow \theta \rangle}$$

$$\frac{s \in H_G \quad id \in H_G(s) \quad \theta = s}{\langle lookup(H_L, H_G, s, id) \rightarrow \theta \rangle}$$

$$\frac{s \in H_L \quad id \notin H_L(s) \quad \theta = lookup(H_L, H_G, H_L(s).par, id)}{\langle lookup(H_L, H_G, s, id) \rightarrow \theta \rangle}$$

$$\frac{s \in H_G \quad id \notin H_G(s) \quad \theta = lookup(H_L, H_G, H_G(s).par, id)}{\langle lookup(H_L, H_G, s, id) \rightarrow \theta \rangle}$$

initialize(schema)
$$\frac{H_L[a \mapsto \{schema.public, par \mapsto \sigma.hf\}] \quad \sigma'._secret \mapsto \{schema.secret\} \quad \sigma'.hf \mapsto a}{initialize(schema) \rightarrow H_L, H_G, \sigma :: \sigma'}$$

return_from_schema
$$\frac{\sigma \equiv \sigma' :: v}{return\_from\_schema \rightarrow H_L, H_G, \sigma'}$$

### B.4.2 SMALL-STEP SEMANTICS

$$\langle H_L, H_G, \sigma, S \rangle \rightarrow \langle H'_L, H'_G, \sigma', S' \rangle$$

$$\text{id} \quad \frac{}{\langle H_L,H_G,\sigma,id\rangle \to \langle H_L,H_G,\sigma,lookup(id)\rangle}$$

$$\text{E.field} \quad \frac{\langle H_L,H_G,\sigma,E\rangle \to \langle H'_L,H'_G,\sigma',V\rangle}{\langle H_L,H_G,\sigma,\text{E.field}\rangle \to \langle H'_L,H'_G,\sigma',\text{get}(V,field)\rangle}$$

$$E_1[E_2] \quad \frac{\langle H_L,H_G,\sigma,E_2\rangle \to \langle H'_L,H'_G,\sigma',V_2\rangle \quad \langle H'_L,H'_G,\sigma',E_1\rangle \to \langle H''_L,H''_G,\sigma'',V_1\rangle}{\langle H_L,H_G,\sigma,E_1[E_2]\rangle \to \langle H'_L,H'_G,\sigma',\text{get}(V_1,V_2)\rangle}$$

$$foo(E_0,E_1,...) \quad \frac{\langle lookup(foo)\to V,V.\_type\equiv Function\rangle \quad \langle H_L,H_G,\sigma,E_0,E_1,...\rangle \to \langle H'_L,H'_G,\sigma',V_0,V_1,...\rangle}{\langle H_L,H_G,\sigma,foo(E_0,E_1,...)\rangle \to \langle H'_L[x_0\mapsto V_0,x_1\mapsto V_1,...],H'_G,\sigma',initialize(V.\_code);V.\_code\rangle}$$

$$E_1[E_2](E_3,E_4,...) \quad \frac{\langle H_L,H_G,\sigma,E_0,E_1,...\rangle \to \langle H'_L,H'_G,\sigma',V_0,V_1,V_2,...\rangle \quad \langle get(V_0,V_1)\to V,V.\_type\equiv Function\rangle}{\langle H_L,H_G,\sigma,foo(E_0,E_1,...)\rangle \to \langle H'_L[x_0\mapsto V_0,x_1\mapsto V_1,...],H'_G,\sigma'[this\mapsto V_0],V.\_code\rangle}$$

$$\text{function}(x_0,x_1,...)\{S\} \quad \frac{}{\langle H_L,H_G,\sigma,\text{function}(x_0,x_1,...)\rangle \to \langle H'_L[a'\mapsto\{...,prototype:a\},a\mapsto],H'_G,\sigma',a'\rangle}$$

$$\text{new foo}(E_0,E_1,...) \quad \frac{\langle lookup(foo)\to V\rangle \quad \langle V.\_type\equiv Class\rangle \quad \langle E_0,E_1,...\rangle \to \langle V_0,V_1,...\rangle}{\langle H_L,H_G,\sigma,\text{new foo}(E_0,E_1,...)\rangle \to \langle H'_L,H'_G,\sigma'[this\mapsto V],init();get(prototype(V),constructor)(V_0,V_1,...)\rangle}$$

$$\frac{\langle lookup(foo)\to V\rangle \quad \langle V.\_type\equiv Function\rangle \quad \langle E_0,E_1,...\rangle \to \langle V_0,V_1,...\rangle}{\langle H_L,H_G,\sigma,\text{new foo}(E_0,E_1,...)\rangle \to \langle H'_L,H'_G,\sigma',V.\_code(V_0,V_1,...)\rangle}$$

$$\{f_1:E_1,f_2:E_2,...\} \quad \frac{\langle H_L,H_G,\sigma,E_1,E_2,...\rangle \to \langle H'_L,H'_G,\sigma',V_1,V_2,...\rangle}{\langle H_L,H_G,\sigma,\{f_1:E_1,f_2:E_2,...\}\rangle \to \langle H_L[a\mapsto\{f_1:V_1,f_2:V_2,...,\_type:object\}],H_G,\sigma,a\rangle}$$

$$(\text{var x = E}) \quad \frac{\langle H_L,H_G,\sigma,E\rangle \to \langle H'_L,H'_G,\sigma',V\rangle \quad \theta=lookup(x) \quad \theta\in H_L \quad \text{fr}=H_L[\theta] \quad \text{fr'}=fr[id\mapsto V]}{\langle H_L,H_G,\sigma,x=E\rangle \to \langle H'_L[\theta\mapsto fr'],H'_G,\sigma',skip\rangle}$$

$$\frac{\langle H_L,H_G,\sigma,E\rangle \to \langle H'_L,H'_G,\sigma',V\rangle \quad \theta=lookup(x) \quad \theta\in H_G \quad \text{fr}=H_G[\theta] \quad \text{fr'}=fr[id\mapsto V\cup fr[id]]}{\langle H_L,H_G,\sigma,x=E\rangle \to \langle H'_L,H'_G[\theta\mapsto fr'],\sigma',skip\rangle}$$

$$(\text{x.f = E}) \quad \frac{lookup(x)\equiv a \quad \theta=H_L(a) \quad \langle H_L,H_G,\sigma,E\rangle \to \langle H'_L,H'_G,\sigma',V\rangle}{\langle H_L,H_G,\sigma,x.f=E\rangle \to \langle H'_L[\theta[f\mapsto V]],H'_G,\sigma',skip\rangle}$$

$$\frac{lookup(x)\equiv \widetilde{a} \quad \widetilde{\theta}=H_G(\widetilde{a}) \quad \langle H_L,H_G,\sigma,E\rangle \to \langle H'_L,H'_G,\sigma',V\rangle}{\langle H_L,H_G,\sigma,x=E\rangle \to \langle H'_L,H'_G[\widetilde{\theta}[f\mapsto V],\sigma',skip\rangle}$$

$$(\text{x[E] = E'}) \quad \frac{lookup(x) \equiv a \quad \theta = H_L(a) \quad \langle H_L, H_G, \sigma, E, E' \rangle \rightarrow \langle H'_L, H'_G, \sigma', V, V' \rangle}{\langle H_L, H_G, \sigma, x[f] = E \rangle \rightarrow \langle H'_L[\theta[V \mapsto V']], H'_G, \sigma', skip \rangle}$$

$$\frac{lookup(x) \equiv \widetilde{a} \quad \widetilde{\theta} = H_G(\widetilde{a}) \quad \langle H_L, H_G, \sigma, E, E' \rangle \rightarrow \langle H'_L, H'_G, \sigma', V, V' \rangle}{\langle H_L, H_G, \sigma, x = E \rangle \rightarrow \langle H'_L, H'_G[\widetilde{\theta}[V \mapsto V']], \sigma', skip \rangle}$$

$$(\text{def foo}(x_0, x_1, ..., x_n)\{\text{Stmt}\}) \quad \frac{\theta = lookup(foo) \quad \theta \in \sigma}{\langle H_L, H_G, \sigma, x[f] = E \rangle \rightarrow \langle H_L[a \mapsto ..., prototype : a', a' \mapsto \{\}], H_G, \sigma[\theta \mapsto a], skip \rangle}$$

$$\frac{\theta = lookup(foo) \quad \theta \in H_L}{\langle H_L, H_G, \sigma, x[f] = E \rangle \rightarrow \langle H_L[a \mapsto ..., prototype : a', a' \mapsto \{\}, \theta \mapsto a], H_G, \sigma, skip \rangle}$$

$$\frac{\theta = lookup(foo) \quad \theta \in H_G}{\langle H_L, H_G, \sigma, x[f] = E \rangle \rightarrow \langle H_L, H_G[a \mapsto ..., prototype : a', a' \mapsto \{\}, \theta \mapsto \theta \cup a], \sigma, skip \rangle}$$

$$(\text{x[E] = E'}) \quad \frac{lookup(x) \equiv a \quad \theta = H_L(a) \quad \langle H_L, H_G, \sigma, E, E' \rangle \rightarrow \langle H'_L, H'_G, \sigma', V, V' \rangle}{\langle H_L, H_G, \sigma, x[f] = E \rangle \rightarrow \langle H'_L[\theta[V \mapsto V']], H'_G, \sigma', skip \rangle}$$

$$\frac{lookup(x) \equiv \widetilde{a} \quad \widetilde{\theta} = H_G(\widetilde{a}) \quad \langle H_L, H_G, \sigma, E, E' \rangle \rightarrow \langle H'_L, H'_G, \sigma', V, V' \rangle}{\langle H_L, H_G, \sigma, x = E \rangle \rightarrow \langle H'_L, H'_G[\widetilde{\theta}[V \mapsto V']], \sigma', skip \rangle}$$

$$\text{if (E) \{ Stmt \}) else \{ Stmt' \}} \quad \frac{\langle H_L, H_G, \sigma, E \rangle \rightarrow \langle H'_L, H'_G, \sigma', False \vee \emptyset \rangle}{\langle H_L, H_G, \sigma, \text{if (E)} \{Stmt\} \text{ else } \{Stmt;\} \rangle \rightarrow \langle H'_L, H'_G, \sigma' \rangle, Stmt}$$

$$\frac{\langle H_L, H_G, \sigma, E \rangle \not\rightarrow \langle H'_L, H'_G, \sigma', False \vee \emptyset \rangle}{\langle H_L, H_G, \sigma, \text{if (E)} \{Stmt\} \text{ else } \{Stmt'\} \rangle \rightarrow \langle H'_L, H'_G, \sigma' \rangle, Stmt'}$$

$$\text{class foo}[M_1, M_2, ..., M_N] \quad \frac{class\_obj = \{M_1, M_2, ..., M_N\}}{\langle H_L, H_G, \sigma, \text{class foo}[M_1, M_2, ..., M_N] \rangle \rightarrow \langle H_L[a \mapsto class\_obj], H_G, \sigma, skip \rangle}$$

$$(\text{return E}) \quad \frac{\langle H_L, H_G, \sigma, E \rangle \rightarrow \langle H'_L, H'_G, \sigma', V \rangle}{\langle H_L, H_G, \sigma, returnE \rangle \rightarrow \langle H'_L, H'_G, \sigma'[returns \mapsto \sigma'[returns] \cup V], skip \rangle}$$

$$\text{for ([let | var] id in E) } \{Stmt\} \quad \frac{\langle H_L, H_G, \sigma, E \rangle \rightarrow \langle H'_L, H'_G, \sigma', V \rangle \quad V.\_proto\_ \equiv \emptyset \quad isEmpty(V) \equiv True}{\langle H_L, H_G, \sigma, \text{for ([let | var] id in E)} \{Stmt\} \rangle \rightarrow \langle H'_L, H'_G, \sigma', skip \rangle}$$

$$\frac{\langle H_L, H_G, \sigma, E \rangle \rightarrow \langle H'_L, H'_G, \sigma', V \rangle \quad V.\_proto\_ \not\equiv \emptyset \quad isEmpty(V) \equiv True}{\langle H_L, H_G, \sigma, \text{for ([let | var] id in E)} \{Stmt\} \rangle \rightarrow \langle H'_L, H'_G, \sigma', \text{for ([let | var] id in V.\_proto\_)} \{Stmt\} \rangle}$$

$$\frac{\langle H_L, H_G, \sigma, E \rangle \rightarrow \langle H'_L, H'_G, \sigma', V \rangle \quad V \equiv X :: V' \quad varDef.type \equiv let}{\langle H_L, H_G, \sigma, \text{for (let id in E)} \{Stmt\} \rangle \rightarrow \langle H''_L, H''_G, \sigma'', \text{initialize(Stmt);let id=X;Stmt; for (let id in V')} \{ Stmt \} \rangle}$$

$$\frac{\langle H_L, H_G, \sigma, E \rangle \rightarrow \langle H'_L, H'_G, \sigma', V \rangle \quad V \equiv X :: V' \quad varDef.type \equiv var}{\langle H_L, H_G, \sigma, \text{for (let id in E)} \{Stmt\} \rangle \rightarrow \langle H'_L, H'_G, \sigma', \text{var id=X;Stmt; for (let id in V')} \{ Stmt \} \rangle}$$

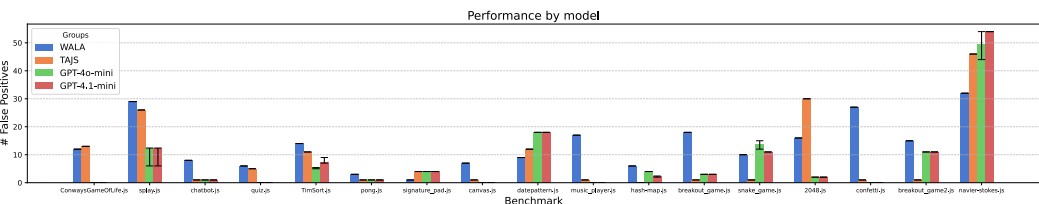

Figure 11: Performance per model on each benchmark program compared to WALA and TAJS.

$$\text{while (E) } \{Stmt\} \quad \frac{\langle H_L,H_G,\sigma,E\rangle \rightarrow \langle H'_L,H'_G,\sigma',V\rangle \quad V\in \text{Falsey}}{\langle H_L,H_G,\sigma,\text{while (E) } \{Stmt\}\rangle \rightarrow \langle H'_L,H'_G,\sigma',skip\rangle}$$

$$\frac{\langle H_L,H_G,\sigma,E\rangle \rightarrow \langle H'_L,H'_G,\sigma',V\rangle \quad V\notin \text{Falsey} \quad \langle H'_L,H'_G,\sigma',\text{Stmt;summarize()}\rangle \rightarrow \langle H'_L,H'_G,\sigma'\rangle}{\langle H_L,H_G,\sigma,\text{while (E) } \{Stmt\}\rangle \rightarrow \langle H'_L,H'_G,\sigma',skip\rangle}$$

$$\frac{\langle H_L,H_G,\sigma,E\rangle \rightarrow \langle H'_L,H'_G,\sigma',V\rangle \quad V\notin \text{Falsey} \quad \langle H'_L,H'_G,\sigma',\text{Stmt;summarize()}\rangle \rightarrow \langle H''_L,H''_G,\sigma''\rangle}{\langle H_L,H_G,\sigma,\text{while (E) } \{Stmt\}\rangle \rightarrow \langle H''_L,H''_G,\sigma'',\text{while (E) } \{Stmt\}\rangle}$$

## C  IMPLEMENTATION AND DATASET

**Implementation.** We implemented ABSINT-AI in 8049 lines of Python, and use Espree brettz9 to parse the Javascript into an AST. We conducted the experiments on a Linux server with two AMD EPYC 7763 64-Core Processors, 128 cores, 1024GB RAM, and 4 NVIDIA RTX 6000 Ada Generation GPUs.

### C.1  DATASET

Table 2: Each program and a small description.

| Program | #Lines | Description |
|---|---|---|
| CGOL.js | 65 | Conway's Game of Life. |
| 2048.js | 234 | The 2048 game implemented for the DOM. |
| breakout_game.js | 158 | An implementation of the Breakout arcade game for the DOM. |
| breakout_game2.js | 91 | A separate implementation of the Breakout arcade game for the DOM. |
| datepattern.js | 91 | Testing date string equality |
| hash-map.js | 577 | A JavaScript implementation of a HashMap. |
| confetti.js | 400 | Confetti animations in the DOM. |
| pong.js | 243 | Pong game in the DOM. |
| snake_game.js | 102 | Snake game in the DOM. |
| books.js | 504 | A library for storing books. |
| FlashSort.js | 84 | Flash Sort. |
| math_sprint.js | 345 | Math calculations in the DOM. |
| drawing-app.js | 442 | A drawing app in the DOM. |
| TimSort.js | 113 | Tim Sort. |
| navier-stokes.js | 385 | Fluid dynamics simulation using a simplified implementation of the Navier–Stokes equations. |
| music_player.js | 196 | Picking between songs to display in the DOM. |
| splay.js | 406 | An implementation of a Splay Tree in JavaScript. |

## D  LLM USAGE

We used a large language model (ChatGPT, GPT-5, OpenAI) to assist with polishing the writing and improving clarity of exposition. The model was not used to design the methodology, conduct

experiments, or generate results. All technical contributions, data analysis, and conclusions are the authors' own.

