# OpenReview forum: "ABSINT-AI: Agentic Heap Abstractions for Abstract Interpretation"
_ICLR.cc/2026/Conference — Submitted to ICLR 2026_

### Official Review · Reviewer_RNoM · 2025-10-29

**Soundness:** 2
**Presentation:** 2
**Contribution:** 3
**Rating:** 2
**Confidence:** 4

**Summary:**

Abstract interpretation is a technique to prove properties of programs. It is a sound approach but incomplete, so it can often fail to prove a property even when the property is true. Such cases are deemed "false positives". One of the central goals in static analysis has been to reduce the number of false positives (that a technique generates), but without compromising soundness (i.e. the number of false negatives should continue to remain zero).

The false positives are a result of abstracting too much. Statically, it is difficult to determine when and how much to abstract. Not abstracting causes non-convergence (of abstract interpretation), but abstracting too much results in increased false positives.

In this paper, the authors propose to use large language models to help with certain choices that an abstract interpreter makes to determine how much to abstract. The main point is that LLMs can take cues from the variable and function names to know what to abstract and what not to. This is a very reasonable hypothesis, and this paper demonstrates that it actually works for "picking heap abstraction" during static analysis of javascript programs.

The paper picks a set of 17 javascript programs on which it compares their approach with baselines that do not use LLMs (and use fixed abstractions). Their approach generates fewer false positives. There are ablations too that show that agentic LLMs do better than just using (non-agentic) LLMs with static prompt.

**Strengths:**

Strengths:
1. The use of language models to help with picking the right abstraction for static analysis is a good and reasonable idea.
2. The experiments clearly show that the proposed approach helps, and alternatives perform worse.

**Weaknesses:**

Weaknesses:
1. While the high-level approach is clear, the details are less clear since the paper lacks explanation on some of the most important bits in the paper.
2. The second weakness is mentioned in the paper -- the approach is limited in scalability. It inherits this non-scalability from static analysis in general. Even the 17 benchmarks had to be carefully chosen (Lines 359 and 362). These are some significant bottlenecks for real use of the approach.

**Questions:**

Questions:
1. All safety violations can be deemed false positives only when the program is known to be safe. How was the safety of the 17 programs established? Was it by manual inspection/verification?
2. Expanding more on the Weakness 1 above, Algorithm 1 provides a good high-level description, but Lines 21 and 22 of Algorithm 1 were not clear. Lines 283-307 is where I expected more details. (I couldn't find those details in the Appendix either).

2.1 First, we have to compute Join at control flow merge points. This happens in loops, but also at if-then-else. So, if we are at if-then-else merge point and doing a Join there, would the "EXEC" action (Loop Execution) make sense there?

2.2 Also, what is meant by "interpreter executes on iteration of the loop" -- do you mean the "abstract interpreter" or is there some concrete "interpreter" also running collecting concrete states?

2.3 The largest program is 570 odd lines. Instead of using "INFO", could we not include the full program in the prompt? In other words, what is the set of "queries" that the agent makes "for program information" on Line 6 of Algorithm 1? Based on Lines 275-- 276, it seems there are two INFO queries: (1) variable inspection and (2) function introspection. Providing the full program can potentially answer (2), and presenting the full abstract state can answer (1) -- these both seem like natural things to provide to the LLM agent? Why not provide them rather than asking the model to query for them? Querying for loop execution seems reasonable.

2.4 Lines 286-303 provide too little detail to make a good judgement on whether the approach is reasonable.

3. Line 427-430: As I mentioned above, when you remove the ability to query and inspect, do you add the "program" and the "abstract state" in full to the context?

Minor typos and errors?:
- line 101: "without affecting soundness" -- I don't think you meant "soundness" there.
- line 134: mentions "even proposing domain-informed widenings" : Widening is tricky because if the LLM proposes a new predicate during widening, the abstract interpreter will not be able to handle that predicate going forward unless it also has abstract transformers for that predicate. Do you really mean "widening" here?
- line 161: "context-sensitive abstraction" -- what is "context-sensitive abstraction" and what is a "context-insensitive abstraction"?
- line 186: "Additional details on our abstract domain...":  Are the abstract domains fixed or are we picking from a collection of abstract domains? In abstract interpretation, if the abstract domain is fixed, then performing analysis on that domain is mostly fixed except for the "widening" step. So, is the LLM being used for widening? Or do we have multiple abstract transformers (for a fixed abstract domain) and we are using the LLM to pick one.
- line 405-408: "Most of the overhead ..." This is also unclear. Out of 500 seconds, 189 are spent on "agent interaction" -- what is "agent interaction" -- is that "querying the agent"? So, 500-189 is on the "abstract interpreter"? But TAJS and WALA abstract interpreters take only 20 seconds? So, even the abstract interpreter is now slower (rather than getting faster because it is presumably being smart about the abstractions it is making)? This is again one of those things that is unclear.
- line 449-450: Did the model have access to the name "conways game of life", maybe it is mentioned somewhere in the code, which helps it propose that abstraction?
- line 462-463: I will be surprised if there is no prior work in program analysis that uses LLM in a way that preserves soundness. In program synthesis, for example, there is plenty of work that uses LLM or ML to only provide heuristics. In theorem proving, LLMs can provide the choice of proof rule, and overall soundness is preserved.

---

> ### Author Response · Authors · 2025-11-18
> **Response to Question 2 (Lack of details on algorithm)**
>
> Thank you for your comments and your thorough reading of the paper! We have added our responses below.
>
> > Algorithm 1 provides a good high-level description, but Lines 21 and 22 of Algorithm 1 were not clear.
>
> Thank you for pointing this out; the behavior was not explained in enough detail in the main text. The LLM acts as a controller over a **fixed set of sound, pre-implemented operators**; it does not modify semantics or generate code. It performs two decision stages:
>
> 1. Selecting which allocation sites to summarize.
>     * At each loop iteration, the interpreter identifies allocation sites whose abstract states changed.
>     * The agent receives a prompt containing:
>         * The loop body and relevant code snippet.
>         * A summary of changed allocation sites (object structures, points-to sets).
>     * The agent can issue a small number of information requests (for example, querying the current abstract value of a variable or requesting a summary of a function’s behavior) or a simulated loop execution with the abstract interpreter.
>     *  Finally, the agent selects a **set of allocation sites to summarize or merge**, ensuring convergence before the next iteration.
> 2. Choosing a merging strategy and widening strategy for each selected site.
>     * For every selected allocation site, the interpreter asks the agent to choose one of the predefined parameterizable merging strategies for that site.
>     * These correspond to standard, sound heap abstractions: allocation-site, recency, and role/field-based merging.
>     * The agent picks among them using natural-language cues from code and variable names.
>
> Every action the agent can take is **predefined, finite, and sound**—it cannot invent new abstractions, only select among existing ones—and all interactions are deterministic within the interpreter. We will add these clarifications to Section 3.3 and the appendix.
>
> >  If we are at if-then-else merge point and doing a Join there, would the "EXEC" action (Loop Execution) make sense there?
>
> The agent is **only invoked at unbounded loop joins**, not at `if`–`then`–`else` merge points. Conditional branches use standard abstract joins and do not require agent intervention. The `EXEC` (loop execution) action is therefore applicable only within iterative contexts where convergence must be ensured.
>
> We will clarify this distinction in Section 3.3 to avoid confusion.
>
> > Also, what is meant by "interpreter executes on iteration of the loop" -- do you mean the "abstract interpreter" or is there some concrete "interpreter" also running collecting concrete states?
>
> “Executing one iteration of the loop” refers to the **abstract interpreter**, not any form of concrete execution. The agent can request that the abstract interpreter perform one additional abstract iteration to expose updated heap and environment states before making its decision.
>
> No concrete program execution or runtime state collection occurs at any point. We will clarify this wording in the revised text.
>
> > The largest program is 570 odd lines. Instead of using "INFO", could we not include the full program in the prompt? In other words, what is the set of "queries" that the agent makes "for program information" on Line 6 of Algorithm 1? Based on Lines 275-- 276, it seems there are two INFO queries: (1) variable inspection and (2) function introspection. Providing the full program can potentially answer (2), and presenting the full abstract state can answer (1)
>
> We experimented with providing the full program and abstract state directly in the prompt. While the code itself typically fits within the context window, the abstract heap and environment representations can grow extremely large—especially for nested loops or deeply structured objects—often exceeding context limits or leading to unstable completions.
>
> To avoid this, we expose the program incrementally through the **INFO queries**. This design keeps context size predictable and ensures that the agent only retrieves the specific information it needs (e.g., variable or function summaries) without overflowing the prompt. In practice, even the changed allocation sites alone can exceed the model’s context when too many iterations are requested, so incremental exposure is necessary for stable, reproducible runs. We will clarify this rationale in the revision.
>
> > Lines 286-303 provide too little detail to make a good judgement on whether the approach is reasonable.
>
> We believe this concern stems from the same ambiguity as above regarding Algorithm 1. Our added description of the two decision stages and example interactions will clarify the reasoning process in Lines 286–303.
>
> > Line 427-430: As I mentioned above, when you remove the ability to query and inspect, do you add the "program" and the "abstract state" in full to the context?
>
> No, we do not. Including the full program and abstract state exceeds the model’s context window in many cases, especially for deeply nested heaps.

---

> > ### Author Response · Authors · 2025-11-18
> > **Response to Weaknesses**
> >
> > > While the high-level approach is clear, the details are less clear since the paper lacks explanation on some of the most important bits in the paper.
> >
> > Our work builds directly on a long line of research in **abstract-interpretation-based heap analysis**, extending these techniques with adaptive, LLM-guided abstraction selection. We recognize that our presentation may not sufficiently explained our core mechanisms, so in the revision we will add additional algorithmic detail to clarify how the agent interacts with the underlying analysis.
> >
> > > The second weakness is mentioned in the paper -- the approach is limited in scalability. It inherits this non-scalability from static analysis in general. Even the 17 benchmarks had to be carefully chosen (Lines 359 and 362). These are some significant bottlenecks for real use of the approach.
> >
> > We agree that scalability is a general limitation of precise JavaScript abstract interpretation. Our approach inherits the same asymptotic behavior as existing analyzers, and our prototype does not target runtime improvements. Our goal is to **increase precision** by enabling adaptive heap abstraction, not to optimize throughput.
> >
> > Regarding the benchmark selection, many constraints arise from the baselines. As noted in Lines 359–362, the main exclusions were due to limitations in **TAJS and WALA**, which do not support several ES2015+ features (e.g., classes, `let`). To ensure fair comparison, we restricted the suite to programs all analyzers could handle.

---

> > > ### Author Response · Authors · 2025-11-18
> > > **Response to typos and errors**
> > >
> > > > line 101: "without affecting soundness" -- I don't think you meant "soundness" there.
> > >
> > > Correct — the intended meaning was *“without affecting precision,”* not soundness. We will fix this wording in the revision.
> > >
> > > > line 134: mentions "even proposing domain-informed widenings" : Widening is tricky because if the LLM proposes a new predicate during widening, the abstract interpreter will not be able to handle that predicate going forward unless it also has abstract transformers for that predicate.
> > >
> > > By “domain-informed widenings,” we do **not** mean that the LLM introduces new predicates or abstract transformers. All widening operators are predefined and sound (e.g., field-set, field-merging, full, depth-based).
> > >
> > > The phrase refers to cases where the agent selects *which fields to widen* based on semantic relevance to the analysis goal. For instance, in a security-oriented analysis, it might preserve fields such as `isAdmin` while widening unrelated ones; in a numerical analysis, it might prioritize keeping numeric fields precise.
> > >
> > > The widening itself remains the standard field-set widening defined in Section 3.3 (Line 300); only the **selection** among existing fields is guided by the agent. We will clarify this wording in the revision.
> > >
> > > > line 161: "context-sensitive abstraction" -- what is "context-sensitive abstraction" and what is a "context-insensitive abstraction"?
> > >
> > > Thank you for pointing this out. We agree that the term *“context-sensitive abstraction”* was imprecise, since “context sensitivity” has a specific meaning in static analysis. Our intent was to describe that traditional analyses apply a **single, global heap-abstraction policy** (e.g., always using recency abstraction), whereas our approach allows the agent to **adapt the abstraction choice per object and per program**, automatically tailoring it to different usage patterns and domains.
> > >
> > > We will revise the wording to avoid confusion with call-context sensitivity—e.g., using *“adaptive abstractions”*—and clarify this distinction in the paper.
> > >
> > > > line 186: "Additional details on our abstract domain...": Are the abstract domains fixed or are we picking from a collection of abstract domains?
> > >
> > > The abstract domain and all abstract transformers are **fixed**. The LLM does not modify the domain, introduce new predicates, or select among multiple abstract transformers. Its role is limited to **deciding when and where to apply widening or merging** within the existing abstract domain.
> > >
> > > In other words, the interpreter implements a single, sound analysis framework; the LLM only guides the *timing and scope* of widening and merging operations, not the semantics of those operations. We will clarify this distinction in the revision.
> > >
> > > > line 405-408: "Most of the overhead ..." This is also unclear. Out of 500 seconds, 189 are spent on "agent interaction" -- what is "agent interaction" -- is that "querying the agent"? So, 500-189 is on the "abstract interpreter"?
> > >
> > > You are correct: the remaining 500 – 189 s corresponds to the **abstract interpreter itself**, which is considerably slower than TAJS and WALA. Much of this difference is due to implementation: both TAJS and WALA are mature, optimized Java analyzers, whereas our prototype is written in **Python**, which introduces substantial overhead.
> > >
> > > Some additional cost may stem from algorithmic differences, since our interpreter maintains a more detailed heap state. However, these algorithmic differences do **not** inflate our reported precision gains: when we disable the LLM and use fixed abstractions within the same interpreter, performance remains much slower than TAJS/WALA but yields comparable precision to their analyses (lines 412-421).
> > >
> > > > line 449-450: Did the model have access to the name "conways game of life", maybe it is mentioned somewhere in the code, which helps it propose that abstraction?
> > >
> > > The program does **not** contain the literal phrase *“Conway’s Game of Life.”* The agent infers the domain from contextual cues such as variable names (`cellState`, `newGeneration`) and control-flow patterns, which strongly indicate the Game-of-Life update logic.
> > >
> > > We view this as an advantage of LLM guidance: it can leverage semantic and naming information— and, if provided, even natural-language comments— to inform abstraction choices that purely symbolic systems cannot exploit. We also observed informally that adding natural-language comments can further improve precision.

---

> > > > ### Author Response · Authors · 2025-11-18
> > > > **Response to Question 1 and missing related work**
> > > >
> > > > > All safety violations can be deemed false positives only when the program is known to be safe. How was the safety of the 17 programs established? Was it by manual inspection/verification?
> > > >
> > > > All 17 benchmarks were **manually inspected** to confirm that the property of interest (absence of unsafe property accesses) holds.
> > > >
> > > > We also note that because our analysis is **sound**, any remaining imprecision manifests only as additional alarms, not missed errors. The correctness of the ground-truth labels therefore does not depend on the behavior of the LLM-guided abstraction.
> > > >
> > > > We will add this clarification to the evaluation section.
> > > >
> > > > > line 462-463: I will be surprised if there is no prior work in program analysis that uses LLM in a way that preserves soundness. In program synthesis, for example, there is plenty of work that uses LLM or ML to only provide heuristics. In theorem proving, LLMs can provide the choice of proof rule, and overall soundness is preserved.
> > > >
> > > > We agree that our work is conceptually related to prior uses of learning and search heuristics within sound reasoning systems. For instance, ML-guided approaches have been used in **compiler optimization selection** [1,2], **proof search and theorem proving** [3]), **program synthesis** [4], and **SAT/SMT solving** [5] to guide heuristic choices while preserving overall soundness.
> > > >
> > > > However, **abstract-interpretation-based program analysis** is its own well-established field, and to our knowledge there has been *no prior work* that integrates LLMs or other ML components into this setting while maintaining soundness. Our approach is the first to do so by constraining the LLM to a fixed, verified set of abstraction operators. We will add this context and a more in-depth related work discussion in the revision.
> > > >
> > > > [1] Ansel, Jason, et al. "Opentuner: An extensible framework for program autotuning." *Proceedings of the 23rd international conference on Parallel architectures and compilation*. 2014.
> > > >
> > > > [2] Huang, Qijing, et al. "Autophase: Compiler phase-ordering for hls with deep reinforcement learning." *2019 IEEE 27th Annual International Symposium on Field-Programmable Custom Computing Machines (FCCM)*. IEEE, 2019.
> > > >
> > > > [3] Bansal, Kshitij, et al. "Holist: An environment for machine learning of higher order logic theorem proving." *International Conference on Machine Learning*. PMLR, 2019.
> > > >
> > > > [4] Li, Yixuan, Julian Parsert, and Elizabeth Polgreen. "Guiding enumerative program synthesis with large language models." *International Conference on Computer Aided Verification*. Cham: Springer Nature Switzerland, 2024.
> > > >
> > > > [5] Ganesh, Vijay, Sanjit A. Seshia, and Somesh Jha. "Machine learning and logic: a new frontier in artificial intelligence." *Formal Methods in System Design* 60.3 (2022): 426-451.

---

> > > > > ### Author Response · Authors · 2025-11-20
> > > > > **Revised Draft**
> > > > >
> > > > > We have revised the draft to address the comments. Updated content in the revision is written in blue text.
> > > > >
> > > > > - Question 1 (safety violations): Line 412
> > > > > - Question 2 (lines 21 and 22 of algorithm 1 unclear): Lines 281-303
> > > > > - Question 2.1 (joins at if-then-else statements): Lines 225-227
> > > > > - Question 2.2 (interpreter execution): line 308, also updated algorithm 1 to explicitly say "Abstract Interpreter" on line 9
> > > > > - Question 2.3 (Including the full program/abstract state): Lines 268-274
> > > > > - Question 2.4 (Missing detail): Lines 281-303
> > > > > - Question 3 (Sending prompt and abstract state when we remove the ability to query): Lines 268-274
> > > > >
> > > > > Typos and errors:
> > > > > - Line 101: "Without affecting soundness" corrected to "Without affecting precision"
> > > > > - Line 134: rephrased to "Widening only fields relevant to the analysis domain"
> > > > > - Line 161: "Context-sensitive abstraction" renamed to "adaptive abstractions"
> > > > > - Line 186: Added elaboration that our abstract domain remains fixed on lines 187-192
> > > > > - Lines 405-408: Added additional detail, now lines 440-444
> > > > > - Lines 462-463: Added additional related works section, lines 512-521
> > > > >
> > > > > We are happy to make any further clarifications or revisions as needed.

---

### Official Review · Reviewer_1rAM · 2025-10-29

**Soundness:** 2
**Presentation:** 3
**Contribution:** 1
**Rating:** 2
**Confidence:** 3

**Summary:**

The paper presents AbsInt-AI, a language-model-guided static analysis framework for JavaScript that augments traditional abstract interpretation with adaptive, per-object heap abstractions. By leveraging an LLM to guide abstraction selection, merging, and widening decisions, the system dynamically tailors its analysis to program context. This approach reduces false positives while preserving soundness in the static analysis process.

**Strengths:**

Using domain knowledge of variable naming conventions, LLMs can provide knowldge that purely symbolic systems cannot infer.The method of using the LLM is not fully described and the results feel incremental.

**Weaknesses:**

The method of using the LLM is not fully described and the results feel incremental. In addition, the paper makes a strong claim of soundness of the resultant analysis but do not provide a proof of this. The LLM is providing hints and assumptions to the analysis that can be incorrect, and thus the outputs can potentially be unsound. The performance is poptentially slow (25x slower than alternatives and requires many iterations). The evaluation is performed on a very small dataset.

**Questions:**

The idea that an LLM can take advantage of naming conventions when doing program analysis and understanding tasks is well accepted and not new. Unfortunately, I couldn't fully understand exactly how the LLM is used to make merge or widening decisions.

The description in Section 3.3 is vague and incomplete. Is the agent just given a menu of strategies to chose from for a particular symbol? This seems arbitrary and it is unclear if this is an optimal strategy, or why this even works.

---

> ### Author Response · Authors · 2025-11-18
>
> Thank you for your comments! We have added our responses below.
>
> > The method of using the LLM is not fully described and the results feel incremental.
>
> Thank you for pointing this out. The LLM’s role is specified on lines 230-240: it selects from a **fixed set of sound, pre-implemented abstraction operators** (e.g., recency merging, allocation-site merging, field-set generalization). These are the only actions the agent can take.
>
> Concretely, during the analysis of unbounded loops the interpreter identifies allocation sites whose abstract states change, presents these and the surrounding code to the agent, and the agent selects which sites to summarize and which merge/widening strategy to apply from this predefined menu. Algorithm 1 formalizes this protocol; we will revise Section 3.3 to make the interaction flow explicit.
>
> Our contribution is not that LLMs understand naming conventions; that is established in prior work. The novelty lies in leveraging the LLMs high-level understanding of the program (including naming conventions) to enable **adaptive, per-object heap abstraction** during analysis—something existing JavaScript analyzers do not support. This adaptivity is what reduces false positives while preserving soundness.
>
> > In addition, the paper makes a strong claim of soundness of the resultant analysis but do not provide a proof of this. The LLM is providing hints and assumptions to the analysis that can be incorrect, and thus the outputs can potentially be unsound.
>
> The analysis does **not** rely on the LLM for soundness. Every abstraction operator the agent may select corresponds to a **pre-established, sound transformation** in the abstract domain (e.g., allocation-site merging, recency abstraction, field-set generalization). These operators are standard in prior work and have known sound transfer functions [1,2,3].
>
> The LLM cannot introduce new rules or assumptions; it can only choose *among these monotone, lattice-ascending operators*. As a result, even an incorrect suggestion cannot make the analysis unsound—it can only reduce precision. (lines 60-65)
>
> We will revise the soundness statement to clarify that soundness derives from the abstract domain and operators, not from the LLM’s correctness.
>
> > The performance is poptentially slow (25x slower than alternatives and requires many iterations).
>
> We agree that introducing the LLM increases runtime. The correct factor, however, is **~9x**, not 25x. The 25x figure compares our unoptimized Python prototype against mature Java implementations (TAJS/WALA). When isolating the LLM’s contribution, only **189 s of 500 s** is due to agent interaction; the remaining overhead is from the Python interpreter itself and our prototype implementation.
>
> Some additional cost may arise from our system’s more detailed heap representation, but this does not inflate our reported precision gains. When the LLM is disabled and the same analysis runs with fixed abstractions, performance remains slower than TAJS/WALA but yields comparable precision. We will clarify these points in the revision.
>
> > The evaluation is performed on a very small dataset.
>
> Our benchmark suite contains **17 programs**, selected to isolate the specific heap–precision issues our method addresses. Real JavaScript programs trigger many other sources of imprecision (e.g., unmodeled builtins, dynamic library behavior) that are **orthogonal to our contribution** and would obscure the effect of heap abstraction choices.
>
> Using small, curated benchmarks is **standard practice for initial static analyzers**: early works such as **TAJS** evaluated on only **8 programs** for the same reason. Proof-of-concept analyzers that introduce new domains or soundness guarantees typically begin with controlled, well-understood suites before scaling to larger corpora. Modeling builtins and external libraries remains an open, independent challenge shared by all analyses.
>
> The suite is drawn from established sources including the **Big Code dataset [4]**, the **V8 benchmark suite**, and selected **NPM libraries**. These programs include both DOM-intensive applications and algorithmic workloads, ensuring coverage of realistic heap-manipulation patterns.
>
> We will add this context to the paper.
>
> [1] Cousot, Patrick, and Radhia Cousot. "Abstract interpretation: a unified lattice model for static analysis of programs by construction or approximation of fixpoints."
>
> [2] Cousot, Patrick, and Radhia Cousot. "Systematic design of program analysis frameworks." *Proceedings of the 6th ACM SIGACT-SIGPLAN symposium on Principles of programming languages*. 1979.
>
> [3] Balakrishnan, Gogul, and Thomas Reps. "Recency-abstraction for heap-allocated storage." *International Static Analysis Symposium*. Berlin, Heidelberg: Springer Berlin Heidelberg, 2006.
>
> [4] Raychev, Veselin, et al. "Learning programs from noisy data." *ACM Sigplan Notices* 51.1 (2016): 761-774.

---

### Official Review · Reviewer_QWjv · 2025-11-01

**Soundness:** 4
**Presentation:** 4
**Contribution:** 4
**Rating:** 8
**Confidence:** 3

**Summary:**

The paper presents an LLM agentic framework for Java script analysis via abstract interpretation. The LLM agent is used to suggest heap abstractions  -- in particular before fixpoint computations in unbounded loops, where the choice of the abstraction influences the performance of the abstract interpreter. The agent inspects the current analysis state, including the heap, code, and abstraction history and selects abstraction strategies from pre-defined possibilities, preserving soundness.

**Strengths:**

The paper presents a clever use of agentic LLMs for sound program analysis for a challenging programming language such as JavaScript.
The use of LLMs and agentic frameworks in this context is new, although the work is similar in spirit with other approaches taht use AI to improve theorem proving.

The paper is well written and a pleasure to read.

The experiments show the merits of the approach compared to traditional symbolic approaches. They also highlight the use of the agentic bit.

**Weaknesses:**

no important weaknesses

**Questions:**

Please comment on the use of such a framework for other programming languages.

---

> ### Author Response · Authors · 2025-11-18
>
> Thank you for your kind comments! We have responded to your question below:
>
> > Please comment on the use of such a framework for other programming languages.
>
> The framework is language-agnostic: the agentic guidance mechanism applies to any language with an abstract interpreter defined over a sound semantic domain. JavaScript presents one of the most challenging test cases due to its highly dynamic heap, so demonstrating success here suggests the approach would transfer readily to more structured languages such as Java.
>
> Adapting to other languages would primarily involve **engineering effort** rather than changes to the underlying methodology. We will note this in the revision.

---

> > ### Comment · Reviewer_QWjv · 2025-11-23
> >
> > Thanks for the response. I remain positive!

---

### Official Review · Reviewer_b4CX · 2025-11-01

**Soundness:** 4
**Presentation:** 4
**Contribution:** 4
**Rating:** 8
**Confidence:** 4

**Summary:**

This paper introduces an agentic framework (called ABSINT-AI)  that assists extends what appear to be sound, essentially conventional static methods by leveraging AI to choose among various heap abstraction approaches at different points in the program. This appears to be a novel way of combining the predictive but unreliable power of LLMs with traditional semantically sound methods for abstract interpretation. The results are promising for a representative analysis problem determining whether a program execution may result in undefined or null dereferences. In comparison with two prior approaches, the method developed in this paper achieves up to a 34% reduction in false positives while maintaining soundness. Further examination shows that the LM’s ability to interact agentically with the analysis environment is crucial, outperforming non-agentic LM predictions by 25%.

**Strengths:**

This is a novel approach, at least to this reviewer. Further. the results are compelling.  It is particularly attractive that the LLM is used only to choose between different abstraction methods; soundess of the analysis does not depend on the LLM.  The paper also seems to open up compelling directions for future work -- can the set of analysis choices be expanded, are there other ways to describe the program setting to the LLM, what are the performance/precision tradeoffs, etc etc.

**Weaknesses:**

WIthin the scope of the paper, there are no evident weaknesses. The work is described adequately, given page constraints, and source code to reproduce these resultrs is given in teh supplementary material.

**Questions:**

Is there additional related work on using LLMs to direct static analysis that should be highlighted for the reader? Perhaps using LLMs to choose proof strategies for verification could be considered analogous.

---

> ### Author Response · Authors · 2025-11-18
>
> Thank you for your comments, we are glad you enjoyed the paper! We have responded to your question below.
>
> > Is there additional related work on using LLMs to direct static analysis that should be highlighted for the reader? Perhaps using LLMs to choose proof strategies for verification could be considered analogous.
>
> We appreciate this suggestion. The most closely related efforts are in **learning-guided formal reasoning**, where LLMs or ML models help select proof rules or strategies within sound systems—for example, in **compiler optimizations** [1,2], **proof search and theorem proving**[3], **program synthesis** [4], and **SMT/SAT solving** [5]. These systems preserve soundness by constraining the model to make choices within verified frameworks, which is conceptually analogous to our setting.
>
> To our knowledge, however, our work is the **first to apply this paradigm to abstract interpretation**, allowing an LLM to guide sound static analysis by selecting among fixed abstraction and widening operators. We will add this discussion to the related work section.
>
> [1] Ansel, Jason, et al. "Opentuner: An extensible framework for program autotuning." *Proceedings of the 23rd international conference on Parallel architectures and compilation*. 2014.
>
> [2] Huang, Qijing, et al. "Autophase: Compiler phase-ordering for hls with deep reinforcement learning." *2019 IEEE 27th Annual International Symposium on Field-Programmable Custom Computing Machines (FCCM)*. IEEE, 2019.
>
> [3] Bansal, Kshitij, et al. "Holist: An environment for machine learning of higher order logic theorem proving." *International Conference on Machine Learning*. PMLR, 2019.
>
> [4] Li, Yixuan, Julian Parsert, and Elizabeth Polgreen. "Guiding enumerative program synthesis with large language models." *International Conference on Computer Aided Verification*. Cham: Springer Nature Switzerland, 2024.
>
> [5] Ganesh, Vijay, Sanjit A. Seshia, and Somesh Jha. "Machine learning and logic: a new frontier in artificial intelligence." *Formal Methods in System Design* 60.3 (2022): 426-451.

---

### Author Response · Authors · 2025-12-03

Thank you to the reviewers and area chairs for the constructive feedback. We summarize below the main points of discussion and revisions to our draft. All updated content in the revision is marked in **blue text**. We are happy to provide any further clarifications if needed.

Reviewers **b4CX**, **QWjv**, and **RNoM** highlighted that the approach is **novel** and demonstrates **compelling precision improvements** over traditional JavaScript analyzers. Reviewer **b4CX** also notes that the work opens promising directions for future extensions of adaptive heap abstraction.

We addressed the following concerns in the revised draft:

- **Soundness:** Clarified our soundness claims and emphasized that the analysis does _not_ rely on the LLM for soundness. (1rAM)
- **Algorithmic Detail:** Expanded Sections **3.2** and **3.3** to provide a detailed, step-by-step walkthrough of the agent’s decision process. (RNoM, 1rAM)
- **Scalability and Benchmarks**: Clarified that scalability follows standard limits of precise abstract interpretation; the 17-program suite isolates heap-precision effects and follows standard practice (e.g., TAJS has 8 programs in its initial benchmark suite). (RNoM, 1rAM)
- **Related Work:** Added a new subsection discussing **LLMs in sound reasoning systems**, including compiler optimization, theorem proving, and SMT/SAT solving. (RNoM, B4CX)
- **Minor Clarifications**: Corrected wording on “soundness vs. precision,” “context-sensitive abstraction,” and clarified that execution refers to the abstract interpreter only. (RNoM)

---

### Meta-Review · Area_Chair_HegE · 2026-01-05

**Summary:**

This paper presents a framework using LLMs to improve static analysis by predicting per-object heap abstractions.  The LLM is used in a controlled fashion, predicting from a fixed set of moves to manipulate the heap (rather like a classifier).  Compared to conventional heap abstraction strategies, an LLM is able to more flexibly reduce false positives while maintaining soundness due to the way it is used.

Strengths

- Clever application of LLMs for static analysis

- Strong empirical results over two existing baselines in this domain

Weaknesses

- Poor presentation of the system design, although this is somewhat addressed.

- Lack of a general ML takeaway: although the authors call this an agent, the nature of the research here is to design a purpose-built agent for static analysis. It is not clear to me what ramifications this paper has for the broader ICLR audience, as this agent does not even generalize to other static analysis tasks. The only real takeaway I see is the broad message that it is generally possible to use LLMs to reason about code in the context of SE/PL systems, which is already established in other literature.

- Non-scalability, both inherited from static analysis in general and also a result of the slowdowns incurred by LLM use

The reviewers are quite divided on this paper. It is stronger as a PL/SE paper than it is as an ML paper, so I will judge it according to that lens.

**Reviewer Concerns:**

Multiple reviewers called out unclear description of LLM use.  I agree with this weakness. The revised version of the paper improves this. However, I still feel that the pseudocode is not as clear as it could be. For instance: "The agent can issue a number of information gathering requests"; I don't really know what this means or how it's implemented. This does seem relevant given that approaches like self-RAG exist for natural language tasks; is this an implementation of those or something different entirely? It feels like a fully-specified implementation might raise additional questions about implementation or ablations.

1rAM

> the paper makes a strong claim of soundness of the resultant analysis but do not provide a proof of this.

This is addressed; it's a feature of how the system is designed.

> slower

Addressed, but the system is still slower.

RNoM

> the approach is limited in scalability. It inherits this non-scalability from static analysis in general.

I don't think this is a totally fair criticism, but the fact that introducing an LLM slows things down is a relevant weakness.

**Reviewer Scores:**

The two higher reviewers would most likely not change their scores.

RNoM: my judgment is that these concerns have been partially addressed through the clarifications of the algorithm.  However, I think new concerns could arise, such as the desire to see an ablation of the INFO criterion.

1rAM: Again, I think the clarifications are addressed, but I don't know if this reviewer would be happy with the answer (the agent does just have a menu of strategies).

---

### Decision · Program_Chairs · 2026-01-26

Reject